# Climate Change Impacts on UNESCO World Heritage-Listed Cultural Properties in the Asia–Pacific Region: A Systematic Review of State of Conservation Reports, 1979–2021

Ky Nam Nguyen [1,2] and Sarah Baker [1,*]

1    School of Humanities, Languages, and Social Science, Griffith University, Gold Coast, QLD 4222, Australia; nam.nguyen2@griffithuni.edu.au
2    Faculty of History, VNU University of Social Sciences and Humanities, Hanoi 100000, Vietnam
*    Correspondence: s.baker@griffith.edu.au

**Abstract:** Utilising the Preferred Reporting Items for Systematic Reviews and Meta-Analyses (PRISMA) 2020 Statement, this article examines 51 UNESCO state of conservation reports from 1979–2021 to explore the impacts of climate change hazards on UNESCO World Heritage-listed cultural properties in the Asia–Pacific region. This article provides a list of the cultural properties impacted by climate change hazards, the types of hazards experienced and their resulting impacts, the kind of conservation responses, challenges to the implementation of the suggested actions, and recommendations for future safeguarding against climate change threats. The article highlights that a negligible number of cultural properties affected by climate change hazards have been monitored and managed by UNESCO, with a much larger proportion of cultural properties facing conservation challenges related to climate change currently going unnoticed by the State of Conservation Information System. Specifically, the review of the state of conservation reports illustrates a very real challenge for understanding climate threats impacting the cultural properties of the Pacific States Parties. While this article confirms the value of the State of Conservation Information System, it also demonstrates that weaknesses remain in its capacity to deliver systematic climate impact analysis.

**Keywords:** Asia–Pacific; climate change; cultural properties; state of conservation; World Heritage; UNESCO

## 1. Introduction

In July 2021, the Henan Province in central China—home to five locations listed as World Heritage Sites by the United Nations Educational, Scientific, and Cultural Organisation (UNESCO)—was inundated by record rainfall and serious flooding. In the city of Anyang in northern Henan, the heaviest rainfall in over 60 years submerged parts of the archaeological site of Yin Xu, located on the banks of the Huanhe River. Inscribed on the UNESCO list in 2006, Yin Xu's outstanding universal value resides in it being one of the earliest identified capital sites in Chinese history "to possess the elements of civilization", conveying in particular the remarkable advancements in science and architectural technology of the late Shang dynasty society [1]. News reports indicated that the entire excavation area of Dasikong village, as well as the remains of the Hunabei handicraft workshops, were submerged, necessitating the evacuation of artefacts, compromising the archaeological pits, impacting relics that could not be moved, and damaging "the 'cultural layers' of human activity recorded in the earth" [2]. Further south on the banks of the Yi River near the city of Luoyang, surging waters contributed to "partial collapses in the soil foundations" of the Longmen Grottoes [2], inscribed on the UNESCO World Heritage list in 2000 because of its stone statues, stupas, and carved inscriptions from the 5th to 8th century [3]. While these cultural works of the Northern Wei and Tang dynasties have an elevated position, which protects them from direct damage, news reports noted that "strong water exposure",

such as that presented by record rainfall and severe flooding, "contributes to long-term corrosion" [2].

The effects on cultural heritage caused by the rain events in Henan, and indeed across many areas of China in 2021, led to warnings that impacts will increasingly be "'supercharged' by climate change", with the National Climate Centre of China stating that "the frequency and severity of extreme temperature and precipitation events will increase in the future" [2]. Flooding is not the only risk to heritage sites as a result of high rainfall events, with damage to World Heritage-listed properties, such as the Mogao Caves in the Gansu Province, also caused by, i.e., fluctuating humidity from precipitation, rainwater seepage, and mudslides [4]. In the context of a climate crisis supercharging the risk to cultural heritage—including the thousand years' worth of Buddhist art preserved in the Mogao Caves, inscribed on the list in 1987—organisations concerned with climate change, such as Greenpeace, have observed that in the last decade, Gansu has experienced higher average temperatures, combined with increased overall total rainfall across fewer days than in the past [5]. Climate change hazards, particularly rain events and associated flooding, have emerged as severe threats to the management, conservation, and promotion of UNESCO World Heritage-listed cultural heritage properties in China and across the Asia–Pacific region, with implications for the structural integrity and authenticity of archaeological remains that have played a critically important role in the history of humanity [6,7].

The ravages of climate change on UNESCO World Heritage-listed cultural properties falls under the remit of the World Heritage Convention—formally known as the Convention Concerning the Protection of the World Cultural and Natural Heritage—adopted by UNESCO in 1972. The Convention's purpose is to ensure that the Outstanding Universal Values (OUV) of cultural and natural heritage are identified, protected, conserved, presented, and transmitted to future generations. It is also the only international convention to safeguard nature and culture by linking them through the concept of heritage [8]. In the UNESCO World Heritage system, there are 1157 listed properties, of which 900 are inscribed in regards to their cultural values (including the aforementioned Yin Xu, Longmen Grottoes, and Mogao Caves sites), 218 on their natural values, and 39 listed as mixed sites, combining cultural and natural qualities. The focus of this article is on the cultural properties which make up nearly 80% of total World Heritage-listed sites. Cultural properties are defined as monuments, groups of buildings, and sites [9] which represent human creativity, values, and cultural traditions and which have played a significant part in human history, settlement, or civilization [10,11]. Approximately 22% of cultural properties (*n* = 195) are located in the Asia–Pacific region.

The impacts of climate change hazards on World Heritage-listed cultural properties in the Asia–Pacific region, along with the management of these impacts, is the focus of this article. According to a recent UNESCO report [12], one in six cultural properties are being threatened by climate change hazards. UNESCO's definition of climate change is borrowed from the United Nations Framework Convention on Climate Change (UNFCCC) [13], as follows: "a change in climate which is attributed directly or indirectly to human activity that alters the composition of the global atmosphere and which is in addition to natural climate variability observed over comparable time periods" (p. 4). In 2008, following the drafting of climate change-focused documents, including "Predicting and Managing the Effects of Climate Change on World Heritage" [14] and "Strategy to Assist Parties to the Convention to Implement Appropriate Management Responses" [14], the World Heritage Convention put forward a standard list of 14 primary factors impacting World Heritage-listed sites, including climate change and severe weather events. Climate change is listed by UNESCO as a general hazard, with associated specific hazards identified as storms, flooding, changes to oceanic waters, droughts, desertification, temperature changes, and other impacts (severe weather events and natural disasters) [6].

The large-scale global impacts of climate change on cultural heritage are now reflected in a growing body of scholarly literature on the matter, as represented in an increasing number of systematic reviews published in the last decade [11,15–19]. These reviews

highlight that studies on climate change and cultural heritage tend to be focused on Europe [11,17,18] and lack international collaboration [18], particularly between scholars from the Global North and South [11]. It can also be observed that studies on climate and heritage emerge from a variety of disciplines in the sciences and social sciences, utilising an array of methods [11,17,18] and while attention to the different climate hazards is uneven [19], there is increasing attention being paid to adaptation and mitigation [15,18] and to climatic problems in historical urban landscape research [16]. Although these reviews focused on different aspects of climate change and cultural heritage, such as barriers to climate adaptation in polar regions [15] or impacts on urban heritage [19], they all share a common reliance on peer-reviewed literature. However, efforts to understand the impacts of climate change on cultural heritage are not restricted to scholarly books and journal articles. Grey literature—that is, materials that have not been published commercially, including for example government reports, practice documents, working papers, green and white papers, technical specifications, discussion papers, program evaluation reports, and standards [20]—is also of value for gauging shifts in the impact of climate change on heritage properties, recommendations for adaptation and mitigation, and management responses. Undertaking a grey literature review can extend the findings from reviews of peer-reviewed literature [21,22], and as a result, grey literature should not be excluded from investigations regarding how climate change is endangering cultural heritage and increasing the challenges to heritage management.

In this article, we turn our attention to the UNESCO state of conservation (SoC) reports as one aspect of grey literature which can be used to better understand the current impact of the climate crisis on World Heritage-listed cultural properties. These reports play a vital role in the UNESCO World Heritage system by providing valuable information on factors affecting property conservation [23]. The World Heritage Committee, for example, draws on these reports to develop specific measures for identified problems, like inscribing a site on the List of World Heritage in Danger [24]. Despite the important role of SoC reports in the World Heritage system, only a small number of scholarly papers have dedicated their attention to the contributions of these reports to provide information that enables the monitoring of threats to property conservation. Meskell [25], for example, provides a robust critique of the World Heritage system through a focus on the politics of heritage management and conservation and how the machinations of nation states can place severe limitations on what becomes recommended or actionable [26]. Despite these challenges, Guzman et al. [23] observe that the SoC reports "represent the most detailed and systematic documentation on heritage at a global scale" (p. 4). With reference to climate change, SoC reports were employed by Guzman et al. [27] to "understand commonalities and differences between the challenges and/or opportunities discussed by conservation practice in all categories of W[orld] H[eritage] properties when confronted by climate change as a global phenomenon of systemic complexity" (p. 5). They note that the value of the SoC reports is their ability to "systematically assess patterns linked to the identification of threats to W[orld] H[eritage] areas, management deficiencies, and conservation needs and developments" [27] (p. 5). However, Guzman et al. [27] observe that while climate change and severe weather events are factors listed as impacting heritage sites, the State of Conservation Information System "does not systematically correlate extreme weather events (e.g., hurricane, drought) with climate change" (p. 5). This serves as a caution for any conclusions that may be drawn from a reliance on SoC reports in regard to understanding the impacts of climate change on cultural properties and their management.

In this article we offer a systematic review of SoC reports attached to World Heritage-listed cultural properties in the Asia–Pacific region to determine how climate change and extreme weather events are presented as a threat to conservation and how climate impacts are remedied. Our review is guided by two key questions: 1. What can be learned about the impacts of climate hazards on cultural properties through an analysis of SoC reports? 2. To what extent can SoC reports deliver systematic climate impact analysis? We narrow our focus to the Asia–Pacific region as, outside of the Europe and North

American regions, it is the World Heritage region identified in the most recent systematic review of scholarly literature by Nguyen and Baker [11] (pp. 2396, 2401), as hosting the cultural properties most impacted by climate change and attracting a significant volume of scholarly interest in that regard. It is also the region in which the authors are based, in terms of their institutional affiliations, and is the location in which they are researching World Heritage-listed properties [28,29]. We begin in Section 2 by outlining the UNESCO State of Conservation Information System before describing in Section 3 the method for data collection and analysis. Section 4 then details the general characteristics of the SoC reports, captured by this article's review process, before highlighting what emerges in the SoC reports in relation to impacts, types of response, challenges, and recommended actions. Sections 5 and 6 reflect on the value of the SoC reports for understanding the impacts of climate change on cultural properties and the management of climate hazards for safeguarding heritage sites, as well as the limitations of the State of Conservation Information System for producing a systematic climate impact and management analysis.

## 2. UNESCO State of Conservation Information System

According to Article 4 of the World Heritage Convention:

"Each State Party to this Convention recognizes that the duty of ensuring the identification, protection, conservation, presentation and transmission to future generations of the cultural and natural heritage. . .situated on its territory, belongs primarily to that State. It will do all it can to this end, to the utmost of its own resources and, where appropriate, with any international assistance and co-operation, in particular, financial, artistic, scientific and technical, which it may be able to obtain." [9]

The World Heritage system supports States Parties in that endeavour through two formal processes—periodic reporting and reactive monitoring [30]. Periodic reporting involves States Parties "provid[ing] an assessment of the[ir] application of the World Heritage Convention" and "whether the Outstanding Universal Value of the properties . . . [are] being maintained over time", as well as "record[ing] the changing circumstances and state of conservation of the properties" [31]. The periodic reporting also offers "a mechanism for regional cooperation and exchange of information and experiences between States Parties concerning the implementation of the Convention and World Heritage conservation" [24,31]. Reactive monitoring, on the other hand, "is the reporting by the World Heritage Centre and the Advisory Bodies to the World Heritage Committee on the state of conservation of specific World Heritage properties that are under threat" [30]. The Convention's Operational Guidelines advise that "States Parties shall submit specific reports and impact studies each time exceptional circumstances occur or work is undertaken which may have an impact on the Outstanding Universal Value of the property or its state of conservation" [24] (p. 56). The SoC reports are part of the reactive monitoring process and present the information received from the State Party, or from other sources, and comments from the State Party and Advisory Bodies for the Committee to consider [32]. Periodic reporting and reactive monitoring have been devised to create a more systematic and standardised assessment of the management of UNESCO World Heritage properties [33]. Between 1979–2021, 4051 SoC reports from 593 properties were prepared as part of the reactive monitoring process [34]. Depending on the number of times the reactive monitoring process was triggered due to threats to a site's conservation, some properties have multiple SoC reports attached, and reflecting the level or persistence of conservation threats, sometimes these occur over consecutive years. More commonly, however, these reports are only requested once from States Parties for a specific property. Of the properties captured in SoC reporting, 150 (25%) are located in the Asia–Pacific region, from 30 States Parties, and 96 of those properties (64%) are cultural properties.

During its annual meeting, the World Heritage Committee and Advisory Bodies, including the International Union for Conservation of Nature (IUCN), the International Council on Monuments and Sites (ICOMOS), and the International Centre for the Study of

the Preservation and Restoration of Cultural Property (ICCROM), discuss the SoC reports and decide on specific types of action. For instance, the committee might request that a State Party address threats through various proposed measures. As part of the reactive monitoring process, States Parties may invite experts to visit the heritage properties to review a conservation issue by way of an advisory mission, or may be required to provide additional information, based on previous decisions of the World Heritage Committee, in regards to the property's state of conservation [35]. In the year following the issue of a SoC report, the committee usually determines whether or not the threats have been appropriately handled. If it is confirmed by subsequent documentation that the threat is being addressed, the World Heritage Committee no longer requires reactive monitoring for the property [33].

At the 32nd session of the World Heritage Committee (2008), the World Heritage Centre was asked to produce an analytical summary of the state of conservation of the World Heritage sites to identify the main threats affecting the properties. An analytical summary was presented at the subsequent session, focusing on SoC reports from 2005–2009 [36]. This document highlighted the need for the systematic monitoring of threats. The World Heritage Centre and the Advisory Bodies had gathered a significant volume of information representing trends in the state of conservation, yet the data were not being managed in a systematic or practical way and thus could not be easily accessed for analytical purposes. UNESCO eventually established the State of Conservation Information System in 2012 as a tool to enable easy access to the reports and decisions of the World Heritage Committee and to provide visualisations of the data using maps, graphs, statistical insights, etc. [37]. With the support of the Flanders government, the State of Conservation Information System was made available to the public, and it is hosted by the World Heritage Centre [37]. While the information is not without its shortcomings—for example, it has been observed that "the capabilities of the system...with regard to search and representation opportunities" would be increased through the incorporation of more extensive "geographical information" [38] (p. 986)—the information system consolidates a significant body of data in one place, enabling ready access by interested parties. As Quesada-Ganuza et al. observe, the UNESCO State of Conservation Information System, and the reactive monitoring process on which it is built, has produced the foundation and conditions to address the impacts of climate change on World Heritage sites [19].

The World Heritage Centre expanded its analysis of conservation trends and challenges in 2014 with its statistical analysis of SoC reports from 1979–2013 [35]. The analysis highlighted various factors affecting World Heritage properties, such as sudden ecological and geological events, local conditions, and invasive species. Importantly, the analysis also captured the impacts of climate change and severe weather events, which had become a focus for UNESCO in 2005 [39], and which were subsequently listed among the 14 factors impacting World Heritage properties [6]. The statistical analysis of the 2642 SoC reports indicated that between 1979–2013, high frequencies of hazards impacting many World Heritage-listed properties included storms ($n = 65$ properties), flooding ($n = 61$), drought ($n = 15$), temperature change ($n = 13$), desertification ($n = 10$), changes to oceanic waters (e.g., water flow, scale, temperature, pH) ($n = 3$), and other climate change impacts ($n = 16$). Notably, storms and flooding were revealed to be major factors, with storms impacting 27% of properties and flooding 25% since 1982 [35]. Regarding the Asia–Pacific region, 561 SoC reports from 111 properties were included in Veillon's statistical analysis [35]. These properties accounted for 50.2% of all the World Heritage-listed sites in the region in 24 States Parties. Of those properties, just over 20% were identified as being impacted by climate change and severe weather events [35].

## 3. Research Materials and Method

This article utilises a systematic review method as a tool to analyse SoC reports. This method is valuable for producing a reproducible structured summary of the field of study [40–42]. It is a method of locating and synthesizing the current state of knowledge

on specific research questions, literature lacunas, and research directions [43]. Systematic reviews have been widely applied to studies on climate change and cultural heritage properties [17]. The systematic review conducted for this article is guided by the Preferred Reporting Items for Systematic Reviews and Meta-Analyses (PRISMA) (see Supplementary Materials) 2020 Statement, which is designed to assist in transparent and robust reporting of the review process through, for example, the provision of flow diagrams that outline the identification, screening, eligibility, and inclusions of sources reviewed [44].

### 3.1. Data Collection

Our review is limited to the state of conservation reports connected to cultural properties in States Parties in the Asia–Pacific region which include qualitative information about climate change hazards. Figure 1 illustrates the number of SoC reports being screened, assessed, and excluded at the different stages of the review process. Using the State of Conservation Information System, it was determined that there were 902 SoC reports from 30 States Parties in the Asia–Pacific region between 1979, the inaugural year of SoC reporting, and 2021, which, at the time of data collection for this review, was the last complete year of reports available in the State of Conservation Information System. The focus on cultural properties reduced the number of reports to 551 from 26 States Parties. The 551 SoC reports were then screened using keywords connected to climate change, including: "climate", "flood", "storm", "drought", "desertification", "ocean", "temperature", "severe weather event", "natural hazard", "typhoon", "cyclone", "hurricane", "mudslide", "landslide", "humidity", "precipitation", and "rain". These keywords were selected based on hazards identified by UNESCO as being climate related and then expanded by drawing on findings from a systematic review of scholarly articles focused on climate change and cultural heritage properties [11].

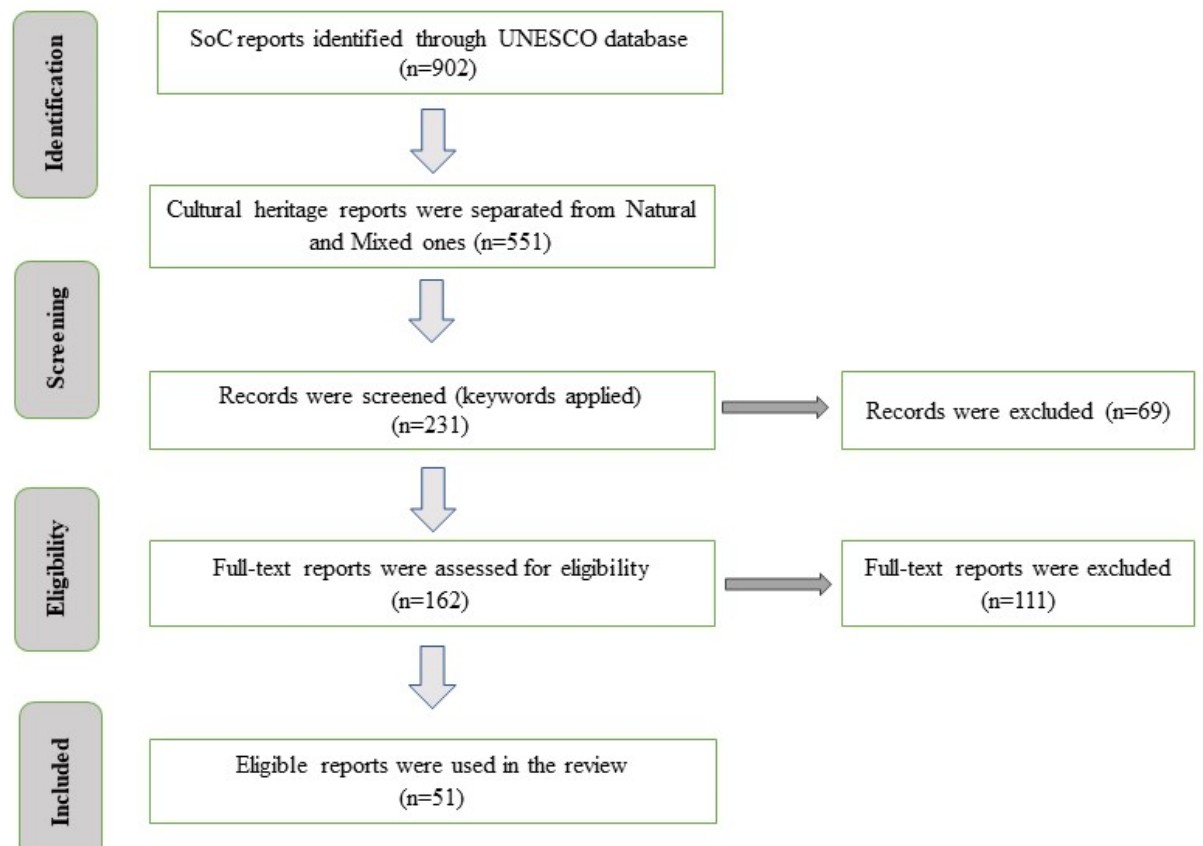

**Figure 1.** Preferred Reporting Items for Systematic Reviews and Meta-Analyses (PRISMA) flow diagram of the selection process of SoC reports, adapted from Moher et al. [45].

Following the application of keywords, 231 SoC reports were selected for further assessment and evaluation. Of those, 69 reports were removed because they mention climate change hazards as factors affecting the property that had been identified only in previous reports or in a general sense. In the case of the Borobudur Temple Compounds in Indonesia, for example, a 2003 SoC report [46] mentions changes to the micro-climate, but only in the context of deforestation due to increasing urbanization, thus leading to the exclusion of the report. The remaining 162 reports were then assessed for eligibility through a close reading of the content. From the close reading, it was determined that an additional 111 reports should be excluded. For instance, natural hazard is mentioned as a factor affecting the Hiroshima Peace Memorial (Genbaku Dome) in Japan, but the earthquakes threatening the property are not connected to climate change in the reports. Similar examples can be found in SoC reports from the Itsukushima Shinto Shrine (Japan), Fujisan, a sacred place and a source of artistic inspiration (Japan), as well as from the Kathmandu Valley (Nepal). Following such exclusions, the PRISMA process resulted in a total of 51 SoC reports which were eligible for review.

*3.2. Material Analysis*

Information from the 51 SoC reports was extracted and compiled in a Microsoft Excel spreadsheet to enable the systematic identification of the main components of the reports. The spreadsheet was designed to capture the general characteristics of the SoC reports, including the year of the report, the name of the States Parties, the name of the cultural properties, and the type of hazards mentioned. A content analysis of the reports was then undertaken, and the content was categorised into various themes. Details added to the spreadsheet at this stage included the impacts of climate hazards on properties, the types of immediate responses, the challenges regarding action, recommendations, and the recommendations for longer-term action.

**4. Research Findings**

This review's findings are represented in two subsections which examine, in turn, the 51 SoC reports' general characteristics and their primary focus. First, we present an analysis of the number of reports according to the year of publication, the number of States Parties captured, the cultural properties included, and the types of climate change hazards experienced. We then detail the information in the reports regarding the impacts of those hazards, the types of responses proposed or taken, the challenges experienced, and the recommendations to address these impacts.

*4.1. General Characteristics*

4.1.1. Year of Reports

While the review encompassed reporting beginning in 1979, the first reference to a climate hazard did not appear until 1991. Figure 2 indicates that the number of SoC reports that include references to climate change grew significantly across the period of 1991–2021. The figure remained relatively low from 1991–2004, with only one report in the years 1991, 1997, 1999, 2000, 2002, and 2004 ($n = 6$). The intervening years registered zero reports, which likely reflects UNESCO's focus on climate change emerging in 2005, with the specified hazards implemented from 2008 on [11]. Between 2004–2017, there was at least one SoC report per year ($n = 29$) which referred to climate change hazards. From 2005–2007, there were 2–3 reports each year ($n = 7$), returning to only one a year from 2008–2010 ($n = 3$). Reports mentioning climatic impacts remained at numbers greater than one per year from 2011–2015 ($n = 13$), with two per year from 2013–2014 ($n = 4$) and three per year from 2011–2012 and in 2015 ($n = 9$). In 2016, the number of reports again fell to one, before escalating to four in 2017. While there were no reports mentioning climate change in 2018 and 2020, the number can still be characterised as representing an upward trend in the reporting of climate hazards impacting cultural properties from 2017–2021 ($n = 21$), with ten reports in 2021 alone. Just over 40% of SoC reports included in the

review appeared between 2017–2021. Visually, the graph demonstrates that climate change hazards are increasingly being referred to in SoC reports as posing significant threats to the conservation of World Heritage-listed cultural properties in the Asia–Pacific region.

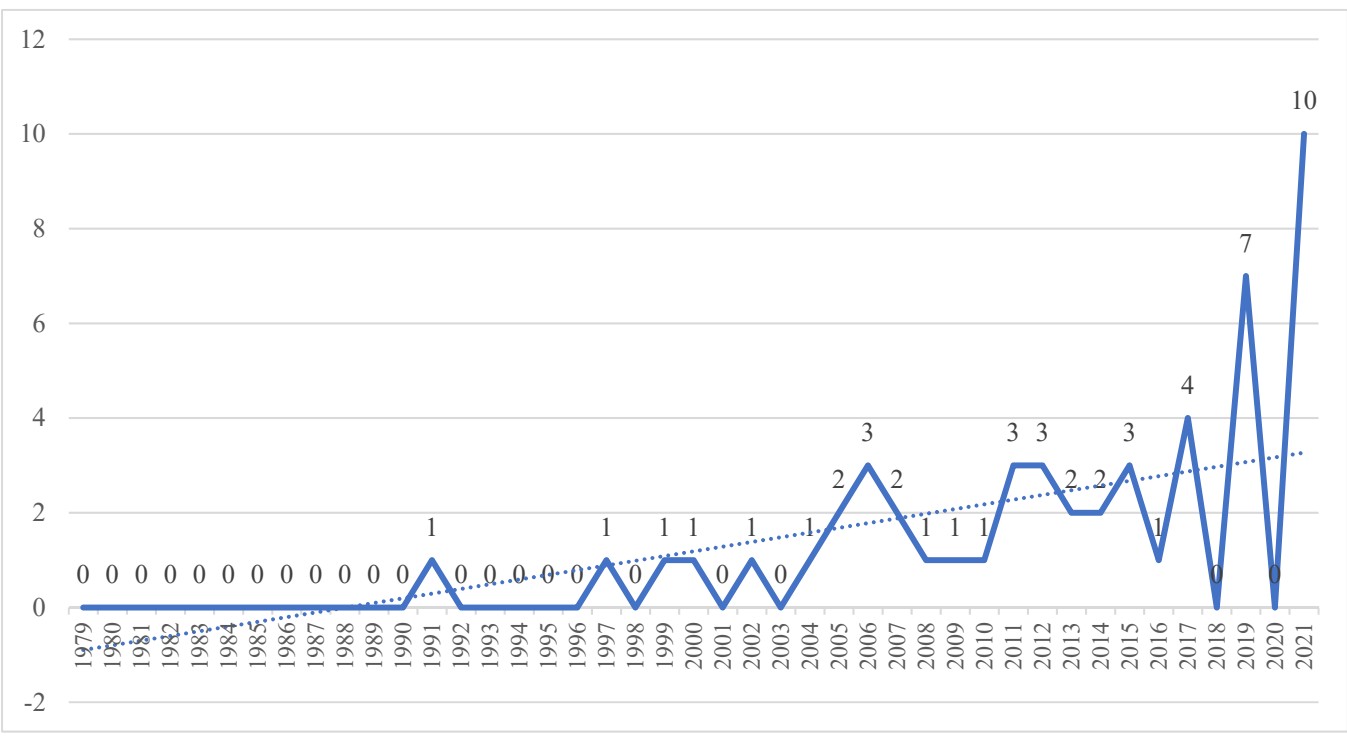

**Figure 2.** Year of SoC reports (1979–2021) in which climate change hazards are referenced.

### 4.1.2. Geographic Distribution

The review revealed 13 States Parties in 19 World Heritage-listed cultural properties with SoC reports that mention climate change hazards (see Table 1). Of these, Federated States of Micronesia was the only State Party from the Pacific captured in the review, with one cultural property having a single SoC report attached. States Parties in Asia, with multiple cultural properties identified as impacted by climate change hazards, included Iran ($n = 4$), China ($n = 2$), India ($n = 2$), and Pakistan ($n = 2$), with the remaining eight States Parties having SoC reports connected to only one cultural property. The States Parties that had triggered the highest number of SoC reports were Pakistan ($n = 11$) and Afghanistan ($n = 10$), followed by the Philippines ($n = 7$), Vietnam ($n = 6$), Iran ($n = 5$), and Thailand ($n = 3$). Cambodia, China, India, Japan, Laos, and Sri Lanka were represented in the review by one SoC report each. The review reveals a significant disparity between the Asia and Pacific areas in terms of the geographic distributions of SoC reports. This may stem from the number of UNESCO cultural properties in Asia ($n = 186$) being much greater than those in the Pacific ($n = 10$). Certainly, within Asia, the review of SoC reports suggests that climate change hazards are impacting cultural properties across a vast geographic area encompassing multiple climate zones.

**Table 1.** SoC reports attached to cultural properties in the Asia–Pacific region which refer to climate change hazards.

| States Parties | Cultural Properties | Year of Inscription | Year of SoC Report | Hazards | World Heritage in Danger Listing |
|---|---|---|---|---|---|
| Afghanistan | Minaret and Archaeological Remains of Jam | 2002 | 2004, 2006, 2008, 2009, 2012, 2013, 2014, 2017, 2019, 2021 | Typhoon, Flooding | 2002–present |
| Cambodia | Angkor | 1992 | 2021 | Flooding | |
| China | Ancient Building Complex in the Wudang Mountains | 1994 | 2014 | Flooding | |
| | The Great Wall | 1987 | 2019 | Flooding | |
| India | Group of Monuments at Hampi | 1986 | 2015 | Flooding | |
| | Mountain Railways of India | 1999 | 2021 | Landslide | |
| Iran | Susa | 2015 | 2017, 2021 | Erosion due to climatic change | |
| | Meidan Emam, Esfahan | 1979 | 2019 | Natural hazard | |
| | Sassanid Archaeological Landscape of Fars Region | 2018 | 2021 | Natural hazard | |
| | The Persian Qanat | 2016 | 2021 | Natural hazard | |
| Japan | Gusuku Sites and Related Properties of the Kingdom of Ryukyu | 2000 | 2021 | Typhoon | |
| Laos | Town of Luang Prabang | 1995 | 2021 | Flooding | |
| Micronesia | Nan Madol: Ceremonial Centre of Eastern Micronesia | 2016 | 2019 | Climate change | 2016–present |
| Pakistan | Archaeological Ruins at Moenjodaro | 1980 | 1991, 2007, 2011 | Flooding | |
| | Historical Monuments at Makli, Thatta | 1981 | 2011, 2012, 2013, 2015, 2016, 2017, 2019, 2021 | Flooding | |
| Philippines | Rice Terraces of the Philippine Cordilleras | 1995 | 2002, 2005, 2006, 2010, 2011, 2012, 2021 | Erosion due to climatic change, Typhoon, Flooding | 2001–2012 |
| Sri Lanka | Rangiri Dambulla Cave Temple | 1991 | 2019 | Natural hazard | |
| Thailand | Historic City of Ayutthaya | 1991 | 2015, 2017, 2019 | Flooding | |
| Vietnam | Complex of Hue Monuments | 1993 | 1997, 1999, 2000, 2005, 2006, 2007 | Typhoon, Flooding | |

### 4.1.3. Types of Climate Change Hazards

Figure 3 illustrates that flooding overwhelmingly appeared in the SoC reports as the climate change hazard with the most widespread impact on cultural properties (*n* = 36 reports, 70%). As Table 1 documents, flooding was represented in reporting related to 11 sites in 9 States Parties. Flooding is mentioned in SoC reports from 1999 onward. However, it is noted that there was a striking increase in the number of reports mentioning flooding in the later period of 2008–2021 (*n* = 25), with 2008 representing the year that climate change hazards were included by UNESCO as a primary hazard impacting World Heritage

properties. Cultural properties located in close proximity to rivers—namely the Complex of Hue Monuments and the Huong (Perfume) River; the Town of Luang Prabang and the Mekong and Nam Khan Rivers; Angkor and the Siem Reap River; the Historic City of Ayutthaya and the Chao Praya, Pa Sak, and LopBuri rivers; the Archaeological Ruins at Moenjodaro and the Indus River; and the Minaret and Archaeological Remains of Jam and the Jam and Hari rivers—appear to be at particular risk of flooding, with flooding highlighted in SoC reports (*n* = 23, 45%) associated with those sites. Figure 4 indicates the causes of flooding at the cultural properties captured in the review, with seasonal flooding appearing most frequently in the SoC reports (*n* = 8), followed by heavy rain (*n* = 3), flash flooding (*n* = 2), heavy rain and snow (*n* = 1), and flooding resulting from changes to river beds (*n* = 1).

　　Typhoons are the second most-common climate change hazard included in the SoC reports (*n* = 6, 12%) for cultural properties in the Asia–Pacific region, but they only appear in relation to three cultural properties attached to the States Parties of Vietnam, the Philippines, and Japan. Properties with multiple reports over a number of years demonstrate that threats can come from several climate related hazards. For example, the Complex of Hue Monuments and the Rice Terraces of the Philippine Cordilleras both feature flooding and typhoons as threats in their SoC reports. Other hazards, such as natural disasters, erosion due to climatic change, landslides, and climate change as a generalised threat, featured in a smaller proportion of SoC reports (*n* = 4, [8%], *n* = 3 [6%], *n* = 1 [2%], and *n* = 1 [2%], respectively).

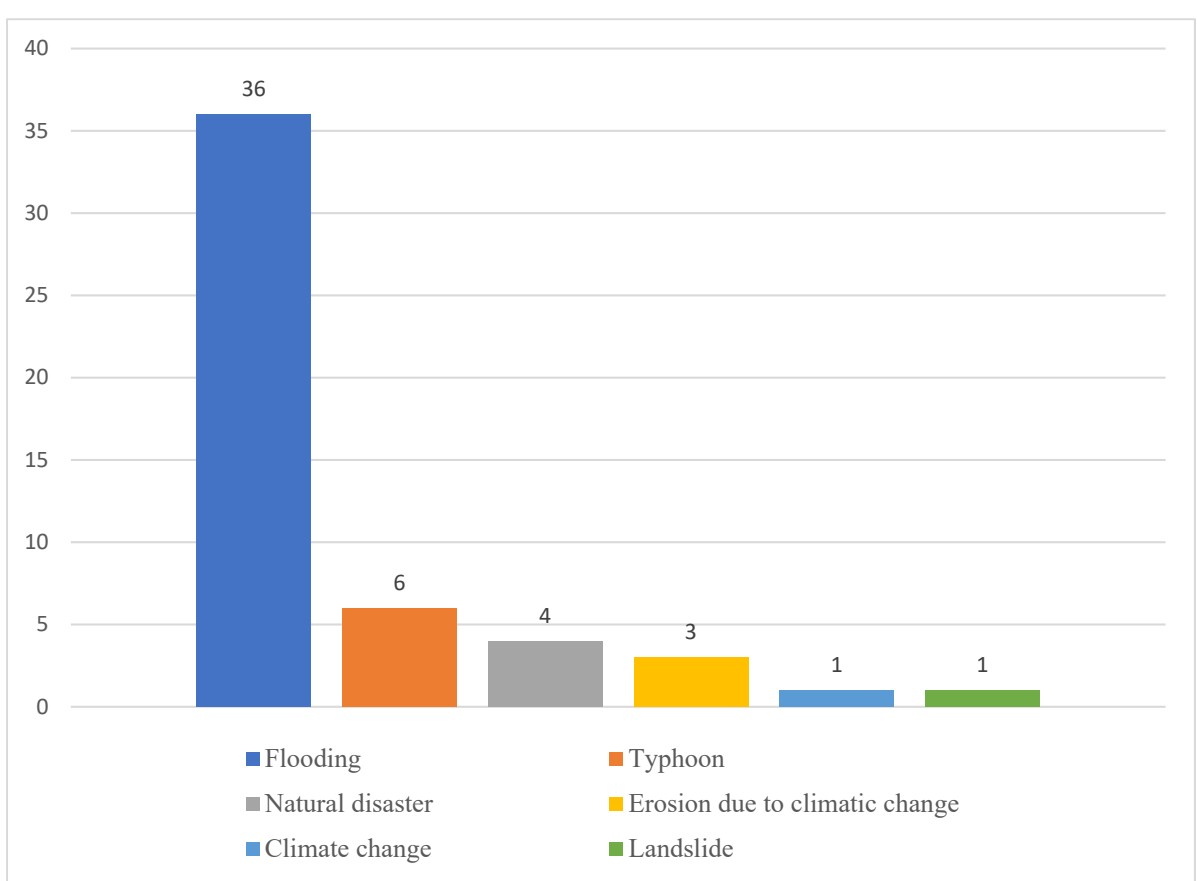

**Figure 3.** Types of climate change hazards featured in SoC reports from the Asia–Pacific region.

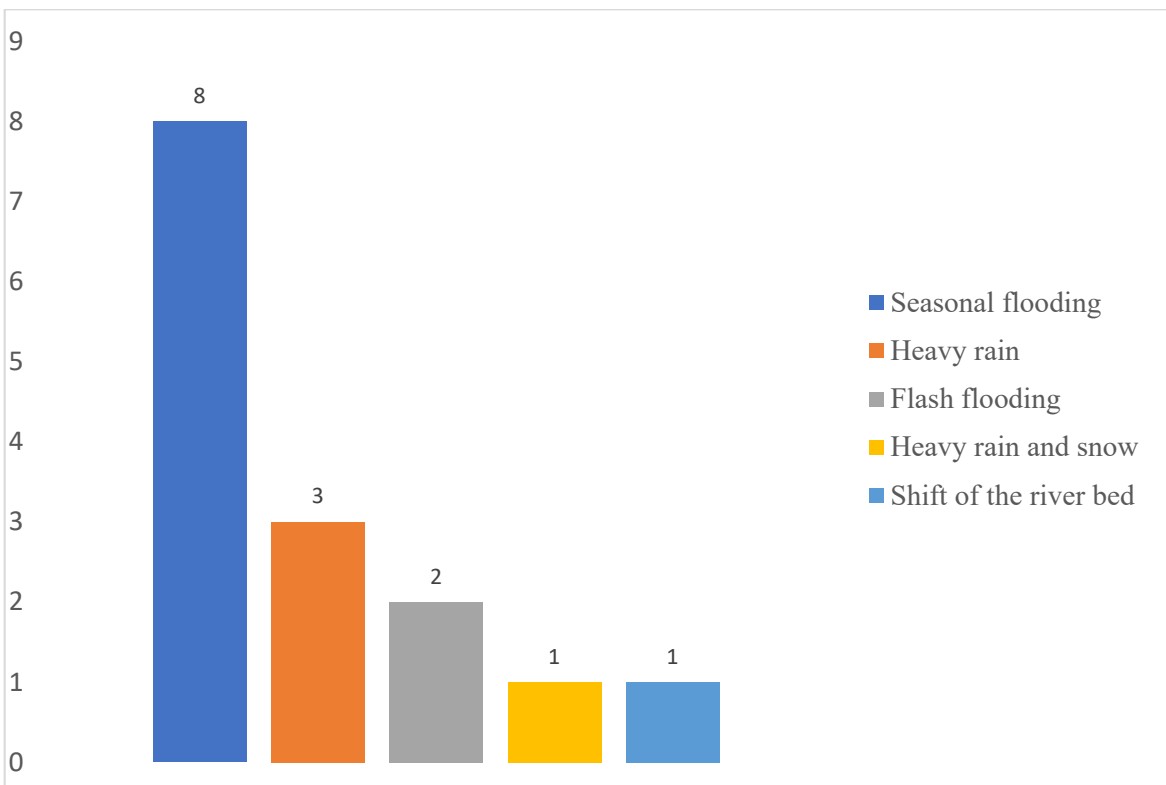

**Figure 4.** Causes of or types of flooding featured in SoC reports concerning cultural properties in the Asia–Pacific region.

It is also noted that three of the properties with SoC reports referencing climate change impacts have been or are currently included on the List of World Heritage in Danger—Minaret and the Archaeological Remains of Jam (Afghanistan), Nan Madol: the Ceremonial Centre of Eastern Micronesia (Micronesia), and the Rice Terraces of the Philippine Cordilleras (Philippines). Climate change hazards are reflected in the in danger listings for these three properties, either in the initial inscription or in the ongoing consideration of retaining that status. The in danger status of the Rice Terraces of the Philippine Cordilleras was, among other things, concerned with climate-induced erosion playing a role in the collapse of terraces, as well as typhoons impacting the irrigation system. When the terraces were removed from the in danger list in 2012, the World Heritage Centre noted that natural disasters would continue to pose a challenge to the conservation of this cultural property [47]. Climate impacts contributing to Nan Madol's in danger status include wave erosion "causing washout" and "undermining the monumental retaining walls in exposed seafront areas", with wave action "forecast to increase in severity due to climate change" [48]. Flooding and erosion are of concern for the conservation of the Minaret and Archaeological Remains of Jam [49].

*4.2. Primary Focus*

4.2.1. Climate Change Impacts

The impacts of climate change hazards recorded in the SoC reports are varied but predominantly refer to extensive damage to historic buildings (e.g., the Complex of Hue Monuments; the Group of Monuments at Hampi; and the Historical Monuments at Makli, Thatta); structural instability (e.g., the Minaret and Archaeological Remains of Jam); damage to key features attributed to the Outstanding Universal Value of a site (e.g., the Rice Terraces of the Philippine Cordilleras); and erosion of structures designed to retain site integrity (e.g., Nan Madol: Ceremonial Centre of Eastern Micronesia). Other impacts connected to climatic change that are highlighted in the SoC reports include long-term damage to

structures from humidity and timber decay, damage to art works (e.g., the Complex of Hue Monuments), and endangerment of peoples' lives (e.g., the Rice Terraces of the Philippine Cordilleras).

### 4.2.2. Types of Response

The SoC reports indicate that there are three types of responses commonly used to deal with climate change hazards: instant response, policy and planning, and traditional knowledge and experience. Instant response includes an emergency support from UNESCO, which can then prompt rescue projects funded and/or delivered by other international organisations (e.g., from Japan, South Korea, the Netherlands, Italy, Switzerland, and the United States of America), as well as additional funding provided by private donors (see e.g., the Complex of Hue Monuments; the Rice Terraces of the Philippine Cordilleras; the Minaret and Archaeological Remains of Jam; and the Group of Monuments at Hampi; Archaeological Ruins at Moenjodaro). Planning and policy measures, on the other hand, are longer-term actions which are designed to tackle the impacts of climatic change through the development of building regulations, preventative projects, flooding monitoring programs, disaster risk management plans, and management frameworks (see e.g., the Complex of Hue Monuments; Angkor; the Minaret and Archaeological Remains of Jam; and the Sassanid Archaeological Landscape of Fars Region). The SoC reports also demonstrate an awareness of the role of traditional knowledge and experience in addressing the impacts of climate change hazards, with reference made to such things as the rehabilitation of traditional water networks (Complex of Hue Monuments), transmission of traditional knowledge and skills in regards to restoration techniques (Rice Terraces of the Philippine Cordilleras), the integration of traditional knowledge systems in education initiatives targeted at improving the state of conservation (Rice Terraces of the Philippine Cordilleras), and exploring traditional knowledge and techniques for flooding mitigation (Historic City of Ayutthaya). Some technical measures are also cited, including the construction of reinforced-concrete retaining walls (Rice Terraces of the Philippine Cordilleras), the creation of flood breakers (Minaret and Archaeological Remains of Jam), and the implementation of survey plans (Susa).

A specific type of response sometimes appears across a number of SoC reports attached to a property over many years of reporting. For instance, the 2005 and 2006 SoC reports for the Complex of Hue Monuments both place an emphasis on a policy and planning response focused on building regulations. This was in response to a prior "inventory of illegal constructions" and "photographic survey", which indicated that following flooding in 1999, "2824 private houses were illegally built on the rampart and bastions of the Citadel, or close to other relevant monuments" [50] and would need to be removed. While not in reference to flooding, these same concerns about houses built without due regard for planning regulations also appear in the SoC reports for the property in 2004, and references to the removal of illegal buildings continues to appear in the SoC reports from 2007, 2009, and 2011. The repeated reference to building regulation in the example of the Complex of Hue Monuments suggests that responses to the impacts of climate change hazards can be ongoing, even if the climate context disappears from the reporting over time.

### 4.2.3. Challenges

Challenges refer to a range of barriers that impede the types of action recommended in the SoC reports. The majority of challenges are related to actions pertaining to policy and planning. Observations are made in the reports in regards to risk management plans not being implemented (Rice Terraces of the Philippine Cordilleras), a fully integrated conservation and management plan not being produced (Sassanid Archaeological Landscape of Fars Region) or the suitability of disaster contingency planning (Archaeological Ruins at Moenjodaro). Challenges are also noted in relation to information being provided by States Parties. For example, in the 2009 and 2012 SoC reports for the Minaret and Archaeological Remains of Jam, the Committee notes that no information has been provided by the State

Party about the fulfilment of corrective measures for flooding. Other challenges included in the SoC reports include the slow progress of management projects (Minaret and Archaeological Remains of Jam), the shortage of sustainable human and financial resources (Rice Terraces of the Philippine Cordilleras), and the ineffectiveness of traditional water networks (Complex of Hue Monuments). Therefore, what the SoC reports suggest is that the ability of States Parties to implement the immediate term actions required by the Committee in order to address climatic impacts on site conservation is regularly hampered by insufficient resourcing. The lack of appropriate resources causes delays in the development of suitable planning documents and the implementation of corrective measures that would address the damage caused by the climate change hazard.

### 4.2.4. Recommendations

The SoC reports also frequently include recommendations geared toward directing States Parties to prepare for climate change hazards that might impact the property in the future. Whereas the "responses" (see Section 4.2.2.) are in relation to taking immediate action to remedy the impacts of a climatic event, the "recommendations" are actions that are future-focused. Recommendations tend to emphasise policy and planning, technical measures, and information and reporting. Regarding policy and planning, a wide range of recommended measures is referred to, including consideration of whether a site should be added to the List of World Heritage in Danger (Complex of Hue Monuments); the implementation of monitoring missions (Historical Monuments at Makli, Thatta; Rice Terraces of the Philippine Cordilleras); the drafting of risk management plans, the updating of site master plans, the development of comprehensive management systems and long-term conservation policies, the implementation of emergency preparedness plans (Susa; Historic City of Ayutthaya; Historical Monuments at Makli, Thatta; Minaret and Archaeological Remains of Jam; Town of Luang Prabang); and the engagement of stakeholders (Gusuku Sites and Related Properties of the Kingdom of Ryukyu). Various technical measures also comprise the recommendations offered by the Committee in the SoC reports. For example, the Afghanistan State Party is asked to begin the systematic collection of data on the Jam and Hari Rivers to help "facilitate planning for future seasonal flooding", as well as to submit a minor boundary modification. Other recommendations include directions related to the mobilisation of international cooperation (Minaret and Archaeological Remains of Jam) and the development of technical reports on the impact of typhoons (Complex of Hue Monuments).

### 5. Discussion

This systematic review confirms that there is value in engaging with SoC reports to gain insight into how climate change is impacting World Heritage-listed cultural properties and the measures being undertaken to address the conservation issues these hazards are creating. Our review of the 51 SoC reports from the Asia–Pacific region which reference climate change hazards highlights that, following an absence of attention to climate matters between 1979–1990, there has been an increase in the reporting of climate impacts in the period of 1991–2021. The type of hazards most commonly reported in the region are flooding, particularly at sites adjacent to rivers, and typhoons. The review also highlights that not all of UNESCO's listed climatic threats are reported as being experienced by cultural properties in the region. For example, absent from the SoC reports captured by the review are hazards like droughts, desertification, and temperature changes. Despite the absence of some hazards, the review demonstrates that States Parties in the region with cultural properties experiencing conservation challenges as a result of climate change are geographically widespread and not confined to a specific climate zone. Flooding, for example, which was revealed to be the hazard most commonly impacting cultural properties in the Asia–Pacific region, is shown to create challenges for the conservation of sites, not only in monsoonal zones, where seasonal flooding might be expected to occur, but also in areas not traditionally impacted by the monsoon season. Climate change hazards

were also shown in the review to be implicated in the listing of three cultural properties previously or currently considered in danger. The article has also highlighted the types of conservation responses, challenges to implementation, and the recommendations for future safe-guarding against climate change hazards. In doing so, it revealed the Committee's emphasis on policy and planning for addressing climate impacts in the immediate and long-term, but also the challenges States Parties face in developing robust planning documents.

Overall, the review has provided a list of the cultural properties impacted by climate change hazards, the types of hazards experienced, and their resulting impacts. The impacts of climate change on cultural properties are consistent with the broader global impacts of climate hazards and natural disasters on different aspects of society and the economy in countries around the world. For example, natural disasters have been shown to lead to changes in patterns of tourism supply and demand [51]. Given that heritage properties are considered a vital tourism resource, climate change impacts on cultural properties, including those in the Asia–Pacific region, may extend to impacts on the flow of tourists to these sites, subsequently impacting economies [52–54]. For example, climatic risks are considered a primary factor impacting the integrity of the Mountain Railways of India. Inscribed in 1999, the railways listing incorporates the Darjeeling Himalayan Railway, the Nilgiri Mountain Railway, and the Kalka Shimla Railway, all of which are fully functional and operational for commuters and tourists. At the time of inscription, it was noted that the railways' personnel are well placed to manage climatic risks to the site, based on a century of experience restoring the line following climatic effects [55]. But past experience does not account for the intensification of climate threats and their possible impact for the future conservation of the railway. The SoC report from 2021 notes that "in August 2020, the State Party confirmed that heavy monsoonal rainfall had induced landslides along the route of the Darjeeling Himalayan Railway" [55]. More recently, news reports observed that the 2023 monsoon season has seen intense flooding impacting the Kalka Shimla Railway, with sections of the line washed away following a landslide, and over 150 disruptions to the operation of the route over a six-week period [56]. Since it is a popular tourist attraction, closures to the Kalka Shimla Railway's operation due to extreme climatic conditions impact social and economic development in the area [56,57]. Generally, the effects of climate hazards on cultural properties like the Mountain Railways of India are not just a matter of damage to built structures and/or intangible heritage, but these effects also impact the heritage engagement and, as a result, the local economies of heritage sites.

### 5.1. The Unnoticed Properties

What is striking is that the number of reports which seriously consider the impacts of climate change hazards on cultural properties is very small ($n = 51$, 9.2%) compared to the overall number of SoC reports attached to cultural properties in the period under review ($n = 551$). While there was an increase in climate change hazard reporting between 1991–2021, the number of references to climatic impacts in the SoC reports is not commensurate with the increasingly acute and omnipresent effects of climate change in the Asia–Pacific region [58], including those on World Heritage-listed cultural properties [11]. What this suggests is that the State of Conservation Information System can only provide a partial and incomplete view of instances of climate change hazards and the ways in which these hazards are handled by States Parties [59]. The review highlights that only a negligible number of cultural properties affected by climate change hazards have been officially monitored and managed between 1979–2021, leading us to propose that a much larger proportion of cultural properties in the region facing conservation challenges related to climate change are presently going unnoticed by the State of Conservation Information System. A case in point is Hoi An Ancient Town, in Vietnam, inscribed on the World Heritage list in 1999. No SoC reports exist for this site, however Hoi An Ancient Town is mentioned in an SoC report attached to the Complex for Hue Monuments [60] as another listed site in Vietnam that was impacted by record flooding [29]. Despite being a site

connected to an extreme flooding event and having a long history of experiencing significant flooding [61], no reactive monitoring process has been triggered for Hoi An Ancient Town.

The State of Conservation Information System cannot, therefore, be relied upon for providing a full picture of the impacts of climate change on World Heritage-listed cultural properties. The recent systematic review by Nguyen and Baker [11], which focused on peer reviewed literature on climate change and cultural properties published between 2008–2021, provides an interesting point of comparison in that regard. The review found that 16 World Heritage-listed cultural properties in the Asia–Pacific region had been reported in scholarly articles as experiencing climate change impacts. Only four of the properties listed by Nguyen and Baker are also captured in this review of SoC reports—Angkor (Cambodia), the Town of Luang Prabang (Laos), the Historic City of Ayutthaya (Thailand), and the Rice Terraces of the Philippine Cordilleras (Philippines). Therefore, in addition to the 19 sites impacted by climate change hazards, with attached SoC reports mentioning climate threats, there are potentially another 12 cultural properties that could have merited the committee's attention due to the impacts of climate change, but for which the reactive monitoring process was never enacted. Those properties and the climate hazards they have experienced are: the Historic Mosque City of Bagerhat (Bangladesh; sea level rise), Lijiang (China; temperature change), the Huashan Cliffs (China; climate change), the Borobudur Temple Compounds (Indonesia; climate change), Bam and its cultural landscape (Iran; flood), the Itsukushima Shinto Shrine (Japan; landslide), Sacred Sites and Pilgrimage Routes in the Kii Mountain Range (Japan; landslide), the Historic Cities of the Straits of Malacca (Malaysia; flood, landslide, sea level rise), Bagan City (Myanmar; flood), Hahoe village (South Korea; flood), Chief Roi Mata's Domain (Vanuatu; storm), and Hoi An Ancient Town (Vietnam; flood). This list indicates that a greater number of States Parties need to tackle the state of conservation of their World Heritage-listed properties in the context of climate change, but they are not benefitting from the reactive monitoring process in order to do so. While a number of those properties have been subject to SoC reporting, those reports did not make reference to the impacts of climate change hazards noted by the scholarly articles captured in Nguyen and Baker's review.

Combining the properties included in this review of SoC reports with those from Nguyen and Baker [11] would still not provide an accurate picture of the full extent of climatic impacts on cultural properties in the region. As Nguyen and Baker [11] observed, their review was limited by its timeframe, sourcing of materials, and restriction to English language publications. Many of the States Parties in the Asia–Pacific region are located in the Global South, where scholars and heritage practitioners encounter structural inequalities that affect different aspects of life, including the capacity to take part in the production and publication of knowledge in the Global North [62]. We might reasonably assume that there is a body of literature in the languages of the States Parties of the Asia–Pacific region which would further reveal cultural properties in which climate change hazards are a significant cause for concern in regards to their state of conservation.

The Missing Pacific

We therefore argue that the impacts of climate change on cultural properties in Asia–Pacific is much greater than reviews of both scholarly literature and SoC reports indicate. In particular, the robust inclusion of cultural properties in the Pacific is missing. The systematic review of SoC reports only located one eligible report from one cultural property from Pacific States Parties (Nan Madol: the Ceremonial Centre of Eastern Micronesia, Federated States of Micronesia). Moreover, the scholarly review by Nguyen and Baker [11] also only listed one cultural property from Pacific States Parties (Chief Roi Mata's Domain, Vanuatu). However, it is widely recognised that the Pacific Islands are particularly vulnerable to the effects of the climate crisis [63,64]), and it would therefore be expected that this vulnerability extends to the impacts of climate change hazards on cultural properties across the Pacific.

Take, for example, Levuka Historical Port Town in Fiji, which does not appear in this review (a single SoC report from 2015 did not mention climate impacts), nor in the

scholarly review of Nguyen and Baker [11], but which is a site where conservation is at significant risk from climate change. These challenges are captured perceptively in an extended passage from the landscape architects and urban designers Penny Allan, Elizabeth Yarina, and Martin Bryant, who describe a visit in 2017 to Levuka Historical Port Town [65] (pp. 109–110):

> The harbour's facilities are less intact: most of the jetties have been washed away. . . . The island's main road, which is still the high street in Levuka, hugs the coastline where it is often inundated when the sea wall is breached by king tides and storms, flooding from upstream, or disrupted by repairs to bridges which cross the numerous creeks. The landscape of the flat lowlands is mostly cleared of vegetation and scored with open channels that don't always deal adequately with extreme rain conditions and often overflow. . . . Following Cyclone Winston [in 2016], many home and business owners in Levuka delayed rebuilding, hoping for UN-designated funds to cover the restoration of heritage buildings as they struggled to meet the demands of reconstruction guidelines. During our visit, a full year later, many of these rebuilding projects continued to languish. Conservation does not happen in a vacuum, and Cyclone Winston is likely to be an early harbinger of increasing climate risk to which Levuka and Ovalau must adapt, as Levuka's historic Beach Street is only a few metres above sea level and floods easily. Whilst the UNESCO evaluation document acknowledges that vulnerability is likely to increase with climate change, what this changing context means for Levuka as a heritage site has not been evaluated. The UNESCO application file lists 'coastal protection and sea buffer boosted' [66], (p. 122) as the long-term strategy to cope with climate change and sea level rise. This response is underdeveloped and fails to address the confluence of runoff with rising seas or increasing cyclone risk. . . . [I]n Levuka when storm surge and sea level rise threaten the integrity of the town's built fabric, the conservation response is to provide immediate protection by raising the level of the sea wall. Ironically this simply increases vulnerability. The wall makes the immediate threat disappear, life goes on as before, people become complacent and have no reason to develop the adaptive strategies that might protect them in the future should the wall fail, or a combination of flooding and storm surge inundate the town.

These authors provide a compelling description of the threat of climate change to this site in the Pacific, and in doing so, they highlight weaknesses in the World Heritage system. It would appear that Levuka is a property that could benefit from the reactive monitoring process in relation to climate change hazards. These benefits relate to the concerted focus on immediate and long-term planning and technical support that SoC reports can offer to better direct conservation efforts on the ground in Fiji. But it also relates to the very thing the State of Conservation Information System is said to provide—a trove of reliable data; the ability to track climate change hazards and their impacts and management over time in cultural properties can only occur if processes are triggered in ways that consolidate data in one place. SoC reporting needs to occur more regularly across the Pacific's cultural properties to ensure that they are represented in the information system in ways that are robust and meaningful and which can better direct efforts to support the conservation of those sites, along with longitudinal analysis of climate impacts and the management response.

## 6. Conclusions

The State of Conservation Information System is a useful tool for evaluating climate change impacts and responses in relation to cultural properties in the Asia–Pacific region. Using the system's dataset, this review has provided a starting point from which to consider the properties being impacted by climate change hazards, the types of impacts being recorded, the immediate and longer-term responses put in place, and the challenges expe-

rienced in implementing those recommendations. However, the capacity for the State of Conservation Information System to provide a robust dataset for an assessment of climate change impacts is challenged by a lack of clarity surrounding the definitions of types of climate change hazards [59], as well as its primary focus on environmental damage to the tangible fabric of sites, with limited regard to social impacts, particularly in urban sites. We also note that our conclusions are limited by the narrow focus of our review on the Asia–Pacific region and the exclusion of natural and mixed properties from the review's purview. In that regard, following UNESCO's initial analysis of SoC reports from 2005–2009 [36] and an extended analysis of SoC reports from 1979–2013 [35], it would be timely for UNESCO to undertake an updated statistical analysis of SoC reports from 2014–2023 in order to provide interested parties with an exhaustive cataloguing of hazards, including those related to climate change, their impacts, and management.

Yet, even if UNESCO were to conduct such an analysis, we would argue that the dataset they are working from is limited by the process through which reactive monitoring occurs, making it difficult in most cases to assess matters of impact and management longitudinally. This is compounded by the brevity of the reports, which often curtails the provision of details that would assist in the monitoring of climate change impacts and conservation responses. For properties for which there are multiple SoC reports over numerous years, while the reports capture the immediate impacts of a significant climatic event, in later reports, they often fail to include the longer-term impacts of climatic change on the property and subsequently, lessen the value of the State of Conservation Information System for longitudinal climate impact analysis.

Despite the challenges inherent in the State of Conservation Information System, it does have a role to play in highlighting climate challenges for cultural properties at a time when a greater understanding of the effects of the climate crisis on built and intangible heritage is needed for States Parties to proactively prepare for adaptation and mitigation of these effects. Although SoC reports are not able to supply systematic insights into interactions between heritage management and climate factors affecting World Heritage properties, in the absence of a more robust system, SoC reports remain a valuable source of information for scholars studying climate impacts on properties inscribed on the UNESCO World Heritage list.

**Supplementary Materials:** The following supporting information can be downloaded at: https://www.mdpi.com/article/10.3390/su151914141/s1.

**Author Contributions:** K.N.N. contributed to the conception, design, acquisition, and analysis of data, as well as drafting, and revising the work to critically guide intellectual contribution. S.B. contributed to supporting the conception and design of the work, auditing the analysis and interpretation of data, and revising the draft critically for intellectual content. All authors have read and agreed to the published version of the manuscript.

**Funding:** This research received no external funding.

**Institutional Review Board Statement:** Not applicable.

**Informed Consent Statement:** Not applicable.

**Data Availability Statement:** The UNESCO State of Conservation reports are available at https://whc.unesco.org/en/soc/?action=list&id_search_region=2, accessed on 1 June 2023.

**Conflicts of Interest:** The authors declare no conflict of interest.

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
