# Peer review of "Climate Change Impacts on UNESCO World Heritage-Listed Cultural Properties in the Asia–Pacific Region: A Systematic Review of State of Conservation Reports, 1979–2021"

_sustainability, doi:10.3390/su151914141_

Round 1
Reviewer 1 Report
Very interesting and well-written article. It explores a type of literature little treated in scientific articles, of enormous interest for its global knowledge.
The choice of the geographical setting is appropriate given the extent, quality and level of risk of the region's heritage assets.
It would be interesting if the authors took into account the following suggestions:
1. In the introduction, a summary of the review articles of the academic literature on natural hazards and heritage should be presented.
2. Devote a few paragraphs to the mechanisms for managing natural hazards (especially those affecting heritage) in the different countries of the Asia-Pacific region.
3. Explain more clearly the system for requesting reports and the need for review or repetition of reports. Table 1 shows properties with several reports, while others have only one or two.
4. Draw a map showing the assets with the highest natural risk due to climate change.
5. Change the type of graph in Figures 3 and 4. I suggest bars ordered from highest to lowest incidence.
6. Define more precisely the terms "Natural Hazard" and "Climate Change" in Figure 3. (They now seem ambiguous. )
7. Introduce into the discussion some ideas about the relationship of the adverse effects of natural hazards on society, especially transport, tourism, etc. In general try to relate it to economic aspects.
Author Response
|
1. In the introduction, a summary of the review articles of the academic literature on natural hazards and heritage should be presented |
This comment is noted, but a summary of the review articles is already made in the article. |
|
2. Devote a few paragraphs to the mechanisms for managing natural hazards (especially those affecting heritage) in the different countries of the Asia-Pacific region |
This comment is noted. However, we already mentioned briefly the relationship between climate change/natural hazards and cultural properties in Asia and the Pacific in the opening sections of the article. The article isn’t about mechanisms for management and devoting three paragraphs to that issue would divert the focus of the article. Mechanisms for management could potentially be the focus of a different article and we will consider that for future work. |
|
3. Explain more clearly the system for requesting reports and the need for review or repetition of reports. Table 1 shows properties with several reports, while others have only one or two |
This comment is well-taken. We have expanded this section to improve clarity of the reporting process. |
|
4. Draw a map showing the assets with the highest natural risk due to climate change |
We noted this comment. However, the article is not concerned with degrees of risk and so it is not clear that there is benefit from adding such a map at this time. Ranking sites based on risk would be a different study and we will consider this for future work. |
|
5. Change the type of graph in Figures 3 and 4. I suggest bars ordered from highest to lowest incidence |
The type of graph in Figure 3 and Figure 4 has been changed accordingly. |
|
6. Define more precisely the terms "Natural Hazard" and "Climate Change" in Figure 3. |
This comment is well-noted. This was a typographical error in the submitted manuscript and the term has now been corrected from “Natural hazards” to “Natural disaster”. |
|
7. Introduce into the discussion some ideas about the relationship of the adverse effects of natural hazards on society, especially transport, tourism, etc. In general try to relate it to economic aspects |
We appreciate this comment. A paragraph has been added in the Discussion section which refers to the impacts of climate hazards on society and tourism, with specific reference to one of the sites in the study’s dataset (Mountain Railways of India). |
Reviewer 2 Report
L.160: remove comment on blind review
L.168: It might be useful to briefly comment on the policy for heritage protection due to climate crisis/change in China and the relevance with the UNESCO policy.
L.198: What is the connection between Periodic/reactive monitoring report and SoCs? A clarification is needed here
L.209: A more thorough discussion on the origins of the Info System is needed, if available, to represent any type of UNESCO policies or politics behind its development.
Author Response
|
Reviewer 2
|
|
|
L.160: remove comment on blind review |
We have removed comment on blind review and replaced with the authors’ references.
|
|
L.168: It might be useful to briefly comment on the policy for heritage protection due to climate crisis/change in China and the relevance with the UNESCO policy. |
We acknowledge this comment. However, the detailed focus on Chinese policy is not aligned with the region-wide focus of the article. We have therefore not added commentary on this issue. |
|
L.198: What is the connection between Periodic/reactive monitoring report and SoCs? A clarification is needed her |
This comment is well-noted and supported by the comment of Reviewer 1. We have revised this section of the article to provide greater clarity of the process, including the connection between Periodic/reactive monitoring report and SoCs. |
|
L.209: A more thorough discussion on the origins of the Info System is needed, if available, to represent any type of UNESCO policies or politics behind its development. |
We have now added additional information about the origins of the Information System. |
Reviewer 3 Report
This study provides a comprehensive and timely analysis of climate change impacts on cultural properties listed by UNESCO in the Asia-Pacific region. By examining 51 State of Conservation (SoC) reports from 1991 to 2021 systematically, the paper investigates the effects of climate change hazards on these cultural properties. The study is valuable for shedding light on UNEDCO's management methods to cope with properties impacted by climate change hazards from various perspectives. Moreover, it highlights the limitations of the State of Conservation Information System in identifying a significant number of cultural properties at risk due to climate change, especially in the Pacific Region. Despite these positive aspects, several major issues deserve attention to enhance the general standard of this article:
1. Regarding the writing of Introduction:
In paragraph three, the authors mentioned that in the UNESCO World Heritage system there are 1157 listed properties, of which 900 are inscribe on their cultural values and 39 for their combination of cultural and natural qualities. However, in the subsequent systematic review of SoC reports concerning nature and cultural properties are excluded from the research materials. It will be more convincing and comprehensive that author explain the reason for doing so, or alternatively, included reports on nature and cultural heritage accordingly.
More recent references concerning the intersection of climate change and cultural heritage should be cited. For example, Tianchen Dai, Xing Zheng, and Jiachuan Yang raised academic awareness of the importance of addressing climatic issues in historical urban landscape studies and incorporating the proper cultural practices from the past into future urban climate management. Their work is titled 'A Systematic Review of Studies at the Intersection of Urban Climate and Historical Urban Landscape' and is published in the Environmental Impact Assessment Review, Volume 97, 2022, with the identifier ISSN 0195-9255. The article can be accessed at https://doi.org/10.1016/j.eiar.2022.106894.
2. Regarding the writing of Material and Method:
In the first paragraph “the article utilises a systematic review method ...is valuable for producing a structured summary of the field of study...”, but the author fails to provide a precise explanation of this method . Suddenly, in Figure 1, the term 'PRISMA flow diagram' is introduced, which can be rather confusing for readers unfamiliar with the acronym. Therefore, I think it’s crucial to clarify that “PRISMA” stand for 'Preferred Reporting Items for Systematic Reviews and Meta-Analyses' and to provide a more detailed literature review of this study method.
Additionally, the article mentions in the Abstract that the span of the State of Conservation (SoC) reports analyzed is '1991-2021.' However, this timeframe isn't further elaborated upon. Notably, the article earlier indicated that SoC reports can be traced back to 1979, which prompts the question: why were reports from 1991 to 2021 selected for analysis? In the second step of PRISMA, the authors employ keywords related to climate change to screen SoC reports. It is important to clarify whether these keywords are established on the foundation of a comprehensive literature review or if they are derived from the authors’ summarization. This distinction is imperative to ensure the credibility and scientific rigor of the study.
3. Regarding the writing of Discussion:
I've noticed that the paper dedicates three paragraphs to elucidate the inadequacies of the current State of Conservation Information System in protecting cultural heritage threatened by climate change. Given the extent of the discussion section and the subsequent context regarding "The Missing Pacific", I would recommend incorporating these paragraphs under a subheading like "The Unnoticed Properties". This could enhance the logical flow of the article.
4. Regarding the writing of Conclusion:
The conclusion feels rather . After the authors demonstrated the weaknesses current system, it would be valuable to discuss potential actions for optimizing the State of Conservation Information System and enhancing future World Heritage Conservation strategies, Furthermore, addressing potential implementation steps or avenues for further academic research would all be worth a few words to provide a well-rounded outline.
5. General issues:
The two pie charts Figure 3 and Figure 4 do not require magnification, instead it would be more beneficial to enhance the resolution and enlarge the fonts of Figure 1.
Author Response
|
Reviewer 3
|
|
|
1. Regarding the writing of Introduction: In paragraph three, the authors mentioned that in the UNESCO World Heritage system there are 1157 listed properties, of which 900 are inscribe on their cultural values and 39 for their combination of cultural and natural qualities. However, in the subsequent systematic review of SoC reports concerning nature and cultural properties are excluded from the research materials. It will be more convincing and comprehensive that author explain the reason for doing so, or alternatively, included reports on nature and cultural heritage accordingly. More recent references concerning the intersection of climate change and cultural heritage should be cited. For example, Tianchen Dai, Xing Zheng, and Jiachuan Yang raised academic awareness of the importance of addressing climatic issues in historical urban landscape studies and incorporating the proper cultural practices from the past into future urban climate management. Their work is titled 'A Systematic Review of Studies at the Intersection of Urban Climate and Historical Urban Landscape' and is published in the Environmental Impact Assessment Review, Volume 97, 2022, with the identifier ISSN 0195-9255. The article can be accessed at https://doi.org/10.1016/j.eiar.2022.106894. |
This comment is well-taken. We have changed expression in the Introduction to better account for the focus on cultural properties only.
We also appreciate the suggestion of the Dai et al reference and it has now been added to the article. |
|
2. Regarding the writing of Material and Method: In the first paragraph “the article utilises a systematic review method ...is valuable for producing a structured summary of the field of study...”, but the author fails to provide a precise explanation of this method. Suddenly, in Figure 1, the term 'PRISMA flow diagram' is introduced, which can be rather confusing for readers unfamiliar with the acronym. Therefore, I think it’s crucial to clarify that “PRISMA” stand for 'Preferred Reporting Items for Systematic Reviews and Meta-Analyses' and to provide a more detailed literature review of this study method. Additionally, the article mentions in the Abstract that the span of the State of Conservation (SoC) reports analyzed is '1991-2021.' However, this timeframe isn't further elaborated upon. Notably, the article earlier indicated that SoC reports can be traced back to 1979, which prompts the question: why were reports from 1991 to 2021 selected for analysis? In the second step of PRISMA, the authors employ keywords related to climate change to screen SoC reports. It is important to clarify whether these keywords are established on the foundation of a comprehensive literature review or if they are derived from the authors’ summarization. This distinction is imperative to ensure the credibility and scientific rigor of the study. |
This comment is well-noted. We have added commentary about systematic review and “PRISMA” to improve clarity.
Thank you for drawing our attention to the discrepancy in dates. We did indeed do a review of reports from 1979 through to 2021, but had dated the review as 1991 because that was the first reference to climate impacts in the SoC reports. We have now amended the date range throughout the article, including the title, and have fixed the Figure so that it now captures the full period.
We have also added an explanation for the start date of 1979 and the end date of 2021.
We now explain the origin of the keywords. |
|
3. Regarding the writing of Discussion: I've noticed that the paper dedicates three paragraphs to elucidate the inadequacies of the current State of Conservation Information System in protecting cultural heritage threatened by climate change. Given the extent of the discussion section and the subsequent context regarding "The Missing Pacific", I would recommend incorporating these paragraphs under a subheading like "The Unnoticed Properties". This could enhance the logical flow of the article. |
We really appreciate this suggestion, and have added a subheading “The Unnoticed Properties”. |
|
4. Regarding the writing of Conclusion: The conclusion feels rather . After the authors demonstrated the weaknesses current system, it would be valuable to discuss potential actions for optimizing the State of Conservation Information System and enhancing future World Heritage Conservation strategies, Furthermore, addressing potential implementation steps or avenues for further academic research would all be worth a few words to provide a well-rounded outline. |
This comment is well-noted. The conclusion has now been expanded, Including discussion of potential actions for the State of Conservation Information System. |
|
5. General issues: The two pie charts Figure 3 and Figure 4 do not require magnification, instead it would be more beneficial to enhance the resolution and enlarge the fonts of Figure 1. |
This comment is well-taken. We changed the type of chart of Figure 3 and Figure 4 and enlarged the font of Figure 1. |
Reviewer 4 Report
Thank you for the opportunity to review the paper and thereby get a glimpse regarding the impacts of climate change on UNESCO World Heritage-listed cultural properties in the Asia and the Pacific Region. The article is interesting, the researched problem has scientific potential, and may be of interest to potential readers. The relevance of the research realized by the authors of the paper is evident. The text of the article is characterized by logicality and the sequence in the presentation of information. I find the topic important from both academic and practitioner perspectives.
Although I evaluate the paper positively, I suggest the authors to make a few more changes in order to improve it:
- The abstract could discuss the findings of the paper and their implications.
- Specific objectives based on the literature review must be clearly presented.
- Hypotheses should be stated and supported by references.
- The theoretical and practical contribution of the paper must be clearly highlighted.
- Shortcomings and limitations of the study with indications for future research should be stated in the conclusion.
- Figures 3 and 4 should be reduced in proportion to the text, taking care not to overlap the words in the printout
- The paper was not prepared in accordance with the journal's instructions, when it comes to in-text citation and the list of references.
Author Response
|
Reviewer 4
|
|
|
- The abstract could discuss the findings of the paper and their implications. |
We appreciate this comment. The original abstract does mention the article’s broad findings and specific implications. So, we have not made changes to the abstract.
|
|
- Specific objectives based on the literature review must be clearly presented. |
The objectives have been added to the paper. |
|
- Hypotheses should be stated and supported by references. |
The research questions have been added to the paper. |
|
- The theoretical and practical contribution of the paper must be clearly highlighted. |
This comment is well-noted. We believe that the theoretical and practical contributions of the article were highlighted in the original manuscript, but note that revisions to the Discussion and Conclusion have now strengthened this. |
|
- Shortcomings and limitations of the study with indications for future research should be stated in the conclusion. |
Limitations are now mentioned in the conclusion. |
|
- Figures 3 and 4 should be reduced in proportion to the text, taking care not to overlap the words in the printout |
We really appreciate this comment. Figures 3 and 4 were changed. |
|
- The paper was not prepared in accordance with the journal's instructions, when it comes to in-text citation and the list of references. |
At this stage, we are using APA in-text citation. The journal’s citation style will be followed once the article is accepted. |